# Trends in the Epidemiology of Allergic Diseases of the Airways in Children Growing Up in an Urban Agglomeration

**DOI:** 10.3390/jcm11082188

**Published:** 2022-04-14

**Authors:** Marcel Mazur, Maria Czarnobilska, Wojciech Dyga, Ewa Czarnobilska

**Affiliations:** 1Department of Clinical and Environmental Allergology, Jagiellonian University Medical College, Botaniczna St. 3, 31-501 Krakow, Poland; marcel.mazur@uj.edu.pl (M.M.); zaklad.alergologii@cm-uj.krakow.pl (W.D.); 2Department of Pathohysiology, Jagiellonian University Medical College, Czysta St. 18, 31-121 Krakow, Poland; maria.czarnobilska@uj.edu.pl

**Keywords:** epidemiology, allergic diseases, asthma, allergic rhinitis, air pollution

## Abstract

The prevalence of asthma and allergies among children has become an increasing problem in the last few decades. Data on the population of children and adolescents, especially living in polluted cities, are limited and based on studies carried out in small groups. In our study, we analyzed the incidence of asthma and allergic rhinitis between 2014 and 2018. We analyzed data collected from nearly 30,000 children aged seven to eight and adolescents aged 16–17, which allowed us to assess the frequency of allergic diseases in the population of children and youth in Krakow. More than 40% of respondents reported allergic symptoms, and nearly 50% of them were not diagnosed and treated. In the group of seven- and eight-year-olds with a positive history of allergies, 52% had allergic rhinitis and 9.1% had asthma. In the group of 16–17-year-olds, allergic rhinitis was diagnosed in 35.8%, while asthma was found in 4.9% of cases. The results obtained over the course of 10 years has shown the reduction in the frequency of asthma (from 22% in the case of asthma in both age groups) and allergic rhinitis cases (from 63.9% in adolescents). In our opinion, this can be considered a positive effect of the preventive measures taken in Kraków after 2010. Although there is still a higher incidence of allergic diseases among children and young people living in urban areas compared to rural areas, the trend apparently has reversed for some diseases.

## 1. Introduction

Allergy is a civilization disease. City inhabitants are particularly vulnerable. According to research from the European Academy of Allergology and Clinical Immunology (EAACI), currently 30% of Europeans have allergy symptoms, with almost half of the cases not diagnosed [1]. In five years, every second European will have some kind of allergy. Allergic diseases have become one of the top three conditions demanding a major effort toward prevention and control in the 21st century, according to the World Health Organization (WHO). The prevalence of allergic diseases among children has become an increasing problem in the last few decades [2,3]. According to data from the 1990s European reviews, the one-year-prevalence rate of bronchial asthma in children varied from 1–3%, when investigated in general practice, to 5–7% in population studies. Eighty percent of the asthmatic children were found to be allergic. The one-year-prevalence rate of rhinitis was 5–10% in general practice, and 10–12% in population studies. About 90% of children with rhinitis symptoms were found to be allergic, with pollen allergy as the most common allergy [4]. In 1997, an ISAAC steering committee studied the prevalence of allergic diseases among children aged 13–14 years in 56 countries, and found that the prevalence of asthma in some Western countries, including the UK and New Zealand, was over 20%, while it was only 2.1% in China [5,6]. In a Korean study from 2010, the prevalence of asthma in children, depending on age, ranged from 9.8% in four- to six-year-olds to 5.4% in 10–13-year-olds, and allergic rhinitis was found to be 38 and 35.9%, respectively [7]. At the same time, the prevalence of asthma symptoms in South America was over 15%, and for rhinoconjunctivitis the prevalence varied from 12.7% for the six- to seven-year-old children to 18.5% for the adolescents [8].

There are no epidemiological studies assessing the development of respiratory allergies in children and adolescents living in contaminated Polish urban agglomerations, including Krakow. Data on the population of children and adolescents in Poland are limited and based on studies carried out in small groups. Moreover, the available studies, as indicated in the discussion below, show epidemiological data from prior to 2010.

In Poland, according to ECAP (Epidemiology of Allergic Diseases in Poland) research conducted in 2006–2008, allergy already affects 40% of Polish society. Twenty-five percent of Poles suffer from allergic rhinitis, and 10% have bronchial asthma [9]. According to epidemiological studies, the number of allergic patients in Poland doubles every 10 years. In our study, conducted as part of the municipal preventive program in Krakow, we analyzed the incidence of asthma and allergic rhinitis in schoolchildren between 2014 and 2018 and the correlation between the proportion of diagnoses of asthma and allergic rhinitis and the concentration of particulate matter in ambient air. The aim of the study was to provide current epidemiological data on the occurrence of asthma and allergic rhinitis in the population of children and adolescents living in the Polish urban areas, determine the percentage of specialist help coverage and to indicate the possible dependence of asthma and allergic rhinitis prevalence on growing up in a polluted environment.

## 2. Materials and Methods

Children and adolescents who participated in the study lived and studied in Krakow. All schools in Krakow were included in the study and all students in the studied age range were given the opportunity to participate in the study. Parents of students who consented to the participation of their children in the study and adolescent participants themselves were provided with questionnaires to be completed via the schools. The questionnaires were distributed to students and then collected by school nurses.

The questionnaire was based on standard questions from the International Study of Asthma and Allergy in Childhood (ISAAC) questionnaire translated from English to Polish by a professional translator, and included questions about attacks of dyspnea, wheezing, coughing and symptoms of non-infectious rhinitis, supplemented with questions about the previous diagnosis and treatment of asthma and/or allergic rhinitis [10]. Therefore, the study, following the pattern of ISAAC, included children from certain age groups: seven- and eight-year-olds and 16 and 17-year-olds.

The survey also included a request for parental (own and parental in the case of adolescents) consent for diagnostic tests, so that in the second part of the program carried out by allergists and allergy nurses, students with a positive allergological history confirmed on the basis of completed questionnaires were referred to allergology outpatient clinics to conduct targeted allergological tests and consultations with an allergist. Based on the questionnaire, supplemented allergological history and physical examination, a preliminary diagnosis was established.

The commissioned allergological diagnostic tests included skin prick tests with inhaled allergens including D. pteronyssinus, D. farinae, grass mix, birch, alder, hazel, mugwort, cat dander, dog dander, Alternaria (Allergopharma, Reinbek, Germany) and, if necessary, spirometry (MES Lungtest 1000, Krakow, Poland).

After analyzing the results of targeted allergy tests a final diagnosis was made at the second consultation visit.

Asthma and allergic rhinitis prevalence was collated with publicly available data from the Chief Inspectorate for Environmental Protection in Poland concerning annual mean concentrations of suspended particulate matter including coarse (PM10) and fine (PM2.5) particles in Krakow from three measurement points [µg/m^3^] in the years 2010–2018.

The study was conducted according to the guidelines of the Declaration of Helsinki, and approved by the Ethics Committee of Jagiellonian University (KBET/23/B/2014).

### Statistical Analyses

All the analyses were performed with Statistica 13 (TIBCO Software Inc., Palo Alto, CA, USA). The significant differences between the groups were calculated using a chi-square test. Trend lines were fitted on the base of linear regression. The Spearman’s test was used to measure the strength of association between two variables. *p* values of <0.05 were considered statistically significant.

## 3. Results

The survey covered 29,872 students in two age groups: seven to eight years (16,623 participants) and 16–17 years (13,249) participants. We found that in the last year surveyed, 45.2% (51.5% in the seven to eight-year-old group, and 32.5% in 16–17-year-olds group) of students were not treated despite the reported symptoms suggesting an allergic disease. The results of the first part of the study are shown in Table 1.

Based on surveys in which at least one positive answer regarding allergy symptoms was marked (7732 children and 4583 adolescents); 2829 participants whose parents declared their intention to continue to participate were qualified for the second part of the study; including 2066 seven to eight year olds and 763 16–17-year-olds.

### 3.1. Results in the Group of Children 7–8 Years

Based on the results of surveys, interviews and allergological tests, children aged seven to eight (2066 participants in the years 2014 to 2018) qualified for additional tests (spirometry, skin prick tests with inhaled allergens).

Spirometry was performed in 680 students. Results indicating forced spirometry and obstructive ventilatory impairment [11] were found in 49 subjects (7.2%).

Skin prick tests with 10 inhalation allergens plus a positive and a negative control were performed in 2038 children reporting respiratory allergy symptoms. At least one positive result for inhalation allergen was found in 766 cases (37.6%). The most common allergens were grass pollen, house dust and flour dust mites, tree pollen, cat allergens, Alternaria spores, mugwort pollen and dog allergens.

A final diagnosis was made in all 2066 students at the second consultation visit.

In the group of seven to eight-year-olds with a positive history of allergies, the following were finally diagnosed: 52% had allergic rhinitis and 9.1% had asthma (Figure 1).

### 3.2. Results in the Group of Adolescents 16–17 Years

Based on the results of surveys, interviews and allergological tests, 763 students aged 16–17 were subjected to additional tests (spirometry, skin prick tests with inhaled allergens) in the years 2014 to 2018.

Spirometry was performed in 252 students. Results indicating forced spirometry and obstructive ventilatory impairment [11] were found in 20 subjects (7.9%).

Skin prick tests with 10 inhalation allergens plus a positive and a negative control were performed in 633 adolescents reporting respiratory allergy symptoms. At least one positive result for inhalation allergens was found in 287 cases (45.3%). The most common allergens were grass pollen, house dust and flour dust mites, tree pollen, cat allergens, Alternaria spores, mugwort pollen and dog allergens.

A final diagnosis was made in all 763 students at the second consultation visit.

In the group of 16–17-year-olds with a positive history of allergy, the following were finally diagnosed: allergic rhinitis in 35.8% (significantly lower compared to the group of seven to eight year-olds *p* < 0.0001), and asthma in 4.9% (*p* = 0.0003) (Figure 1).

The performed statistical analysis (chi-square test) allowed us to confirm the trend of decreasing frequency of diagnoses of asthma and allergic rhinitis in both studied age groups over the years 2014–2018 (Figure 2 and Figure 3).

Moreover, a strong correlation was found between the frequency of diagnoses and the decreasing concentration of PM10 in the air from the Krakow measuring station located in in the very center of the city (Krasińskiego Street), which was statistically significant for asthma in both age groups—*p* = 0.013 and 0.005, respectively, and allergic rhinitis in the younger group—*p* = 0.025. The statistical significance of the correlation with the concentration of PM10 in the older age group was found for another measuring station (Bulwarowa Street)—*p* = 0.033.

There was also a correlation between the proportion of diagnoses of asthma and allergic rhinitis and the concentration of PM10 and PM2.5 at other measuring stations, but with the exception of the correlation between the prevalence of allergic rhinitis in the older age group and the concentration of PM2.5 for the Krasińkiego Street measuring station, they were not statistically significant (Table 2).

## 4. Discussion

More than 20 years have passed since the first Polish epidemiological studies on the prevalence of asthma and allergic rhinitis in the population of children and adolescents. In the Polish Multicenter Study of the Epidemiology of Allergic Diseases (PMSEAD) conducted in 1998–1999, asthma was found in 8.6% and seasonal allergic rhinitis in 8.6% (persistent 2.1%) of children aged 3–16 years [12].

Research conducted over 10 years later (Epidemiology of allergic diseases in Poland, pol. Epidemiologia Chorób Alergicznych w Polsce, ECAP) in 2006–2008 for urban areas in Poland showed that asthma affected 4.4% of children aged six to seven years and 6.5% of children aged 13–14 years [9]. For allergic rhinitis, it was 24.3 and 25.2%, respectively.

In a more recent study published in 2016 (The Belarus, Ukraine, Poland Asthma Study, BUPAS), the prevalence of asthma in children 7–13 years of age in Poland in urban areas was 4.1% [13].

It is worth noting that the protective effect of farm-living on the prevalence of atopy and overall allergic diseases in children in Poland is still observed [14,15,16]. Urban children have a higher overall prevalence of allergic diseases and atopy than children living in rural areas [17]. A possible explanation is the influence of air pollution on the development of allergic diseases [18]. Given the scale of the problem, it is essential to implement measures for prevention and early diagnosis of allergic disorders to minimize their distant health effects [16].

The presented survey conducted as part of the preventive program showed that almost 40% of Krakow’s schoolchildren have allergic symptoms and 45% are not treated for them. If these diseases are not recognized and not treated as soon as their symptoms appear they naturally undergo further development, leading to a significant drop in quality of life and disability. Most allergic conditions start in childhood and disproportionately affect children and teenagers, making the chance of a severe allergic reaction during school hours high, yet school personnel are often not informed or prepared for such events. Early detection of the etiological factors of an allergic disease allows for an expert opinion and thus an interruption in the development of the disease, avoiding allergic reactions associated with the risk of life-threatening acute asthma measures or anaphylactic shock as well as financial expenditures on treatment.

The Polish part of The International Study of Asthma and Allergies in Childhood (ISAAC) study assessed changes in asthma prevalence between 1994/1995 (study I) and 2001/2002 (study II) in populations of school children of six to seven years of age and 13–14 years of age surveyed in one of the largest Polish cities (Krakow and Poznan). The prevalence of the asthma symptoms and established asthma diagnosis increased in those seven years; in the six to seven years of age group in Krakow it increased from 4.0 to 5.8%, in Poznan from 1.3 to 5.9%; in the 13–14 years of age group in Krakow it increased from 2.3 to 6.8%, while in Poznan increased from 2.0 to 5.2%. The authors noted that a significant number of children presenting asthma symptoms remained underdiagnosed [19].

As the municipal prevention program was also conducted in 2006–2009 [20,21], we were able to compare the data obtained over 10 years, i.e., in 2008 and 2018, in order to have an insight into the trend of epidemiology of allergic diseases in the population of Krakow children and adolescents over 10 years.

In 2008, in a group of seven to eight-year-olds, 51.2% had allergic rhinitis and 22.0% had asthma (no change for allergic rhinitis and 13% less for asthma compared to 2018 data). In the group of 16–17-year-olds allergic rhinitis was diagnosed in 63.9% while asthma was diagnosed in 22.0% of cases, (28% less for allergic rhinitis and 17% less for asthma compared to 2018 data).

Comparing the results obtained over the course of 10 years, a reduction in the prevalence of asthma and allergic rhinitis (in adolescents) can be seen. In our opinion, this can be considered as a positive effect of the preventive measures taken at that time. Thanks to the implementation of the municipal preventive program, we showed how many children and young people in Krakow struggle with allergic diseases. According to the data on air quality in Kraków, the concentration of PM10 and PM2.5 gradually decreased over the years 2010–2018 (Figure 4), showing a clear downward trend [22]. Also, the fact that air pollution can be a contributing factor [23,24] has caused children and their parents to be aware of the need to avoid exposure to polluted air and to wear face masks. Less dust reaching the respiratory tract means fewer factors provoking the development of inflammation in the nose and bronchi. This may explain why there was a downward trend of asthma and allergic rhinitis incidence in 2014–2018 and also why the incidence of asthma in children and adolescents has decreased twice over the ten year period between 2008 and 2018.

## 5. Conclusions

In conclusion, although there is still a higher incidence of allergic diseases among children and young people living in urban areas compared to rural areas, as shown in previous research conducted in Poland, the trend apparently has reversed for some diseases. In our study, we were able to show a reduction in the incidence of asthma and allergic rhinitis over the 10 years analyzed, which can be associated with an improvement in air quality in the city where the participants live. If we could use this knowledge, we could count on slowing or even stopping the increasing numbers of allergic diseases among children and adolescents by continuing efforts to improve ambient air quality.

## Figures and Tables

**Figure 1 jcm-11-02188-f001:**
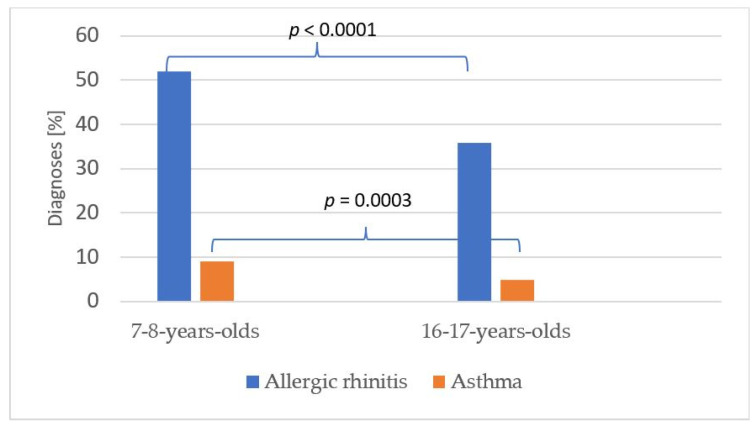
Percentage of final diagnoses in examined groups (seven to eight years, 2066 patients and 16–17 years, 763 patients).

**Figure 2 jcm-11-02188-f002:**
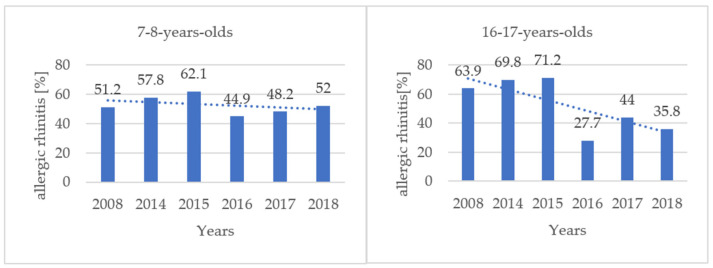
Incidence of allergic rhinitis in examined groups (seven to eight years, 2066 patients, regression coefficient = −0.120; *p* = 0.879 and 16–17 years, 763 patients, regression coefficient = −0.571; *p* = 0.236) in 2008–2018 (2009–2013 non tested).

**Figure 3 jcm-11-02188-f003:**
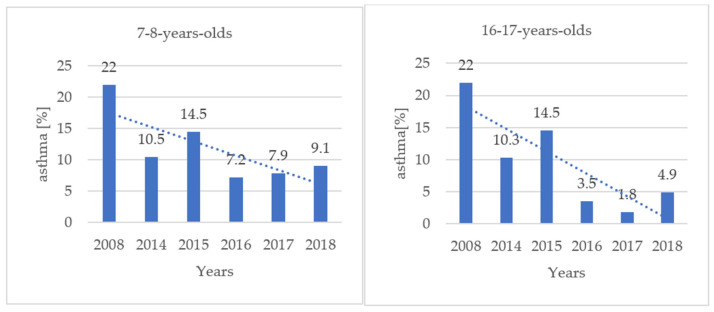
Incidence of asthma in examined groups (seven to eight years, 2066 patients regression coefficient = −0.908 *p* = 0.012 and 16–17 years, 763 patients, regression coefficient = −0.898; *p* = 0.015) in 2008–2018 (2009–2013 non tested).

**Figure 4 jcm-11-02188-f004:**
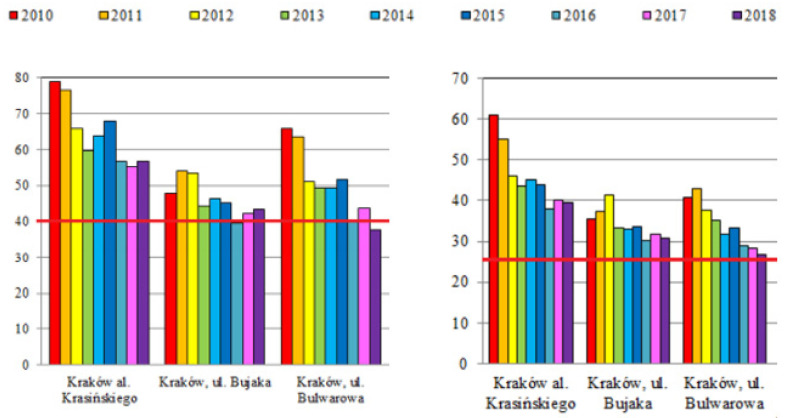
Annual mean concentrations of PM10 (**left**) and PM2.5 (**right**) in Krakow from three measurement points [µg/m^3^] in the years 2010–2018 according to Chief Inspectorate for Environmental Protection in Poland (modified); the norm value (40 µg/m^3^ for PM10 and 25 µg/m^3^ for PM2.5) is marked with a red line.

**Table 1 jcm-11-02188-t001:** The studied group of students.

Group	Survey Year	7–8 Years	16–17 Years
Students included in the survey	2018	3762	2651
	2017	2634	2516
	2016	2891	2628
	2015	3299	1997
	2014	4037	3457
Students reporting symptoms suggesting an allergic disease/percentage	2018	1591/42.3	793/29.9
	2017	939/35.6	662/26.3
	2016	1240/42.9	1212/46.1
	2015	1557/47.2	629/31.5
	2014	2396/59.4	1287/37.2

**Table 2 jcm-11-02188-t002:** Analysis of the correlation between the diagnosis of asthma and allergic rhinitis and the concentration of PM 10 and PM2.5 in the studied years.

Correlation Coefficient	Asthma 7–8	Asthma 16–17	AR 7–8	AR 16–17
PM10 Krasinskiego	0.950; *p* = 0.013	0.973; *p* = 0.005	0.923; *p* = 0.025	0.688; *p* = 0.199
PM10 Bujaka	0.549; *p* = 0.338	0.657; *p* = 0.228	0.465; *p* = 0.430	−0.007; *p* = 0.992
PM10 Bulwarowa	0.839; *p* = 0.075	0.870; *p* = 0.055	0.851; *p* = 0.068	0.908; *p* = 0.033
PM2.5 Krasinskiego	0.693; *p* = 0.194	0.857; *p* = 0.064	0.846; *p* = 0.071	0.898; *p* = 0.038
PM2.5 Bujaka	0.221; *p* = 0.721	0.341; *p* = 0.514	0.084; *p* = 0.893	−0.212; *p* = 0.732
PM2.5 Bulwarowa	0.775; *p* = 0.124	0.826; *p* = 0.085	0.722; *p* = 0.169	0.712; *p* = 0.177

## Data Availability

Publicly available datasets were analyzed in this study. This data can be found here http://powietrze.gios.gov.pl/pjp/documents/download/103138 (accessed on 18 December 2021).

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
