# Peer review of "Trends in the Epidemiology of Allergic Diseases of the Airways in Children Growing Up in an Urban Agglomeration"

_jcm, 2022, doi:10.3390/jcm11082188_

Round 1

Reviewer 1 Report

Compared to the previous version of the manuscript, the authors have made some changes resulting in improved quality of the manuscript; however there are still several issues to solve:

  • The Introduction section could be expanded with more data on the epidemiology of allergic diseases (for example in other European cities, rural vs. urban areas, in the world etc.)
  • How did the authors ensure that the study sample was representative for the target population?
  • The Results section should be improved as follows:
    • the authors could provide an estimation of the total number of children in the target population
    • it is not clear in the text: how many students actually qualified for the second part of the study (both in 7-8 and 16-17 years age group)? There may have been students who qualified based on the answers to the questionnaire but their parents or themselves did not consent for further testing. How many were in this situation? The numbers should be provided
    • Among 2066 children in the 7-8 years age group referred for allergy testing, only 2038 were actually tested. The authors should explain why. Similar comment for the 16-17 years age group.
    • The provided number of children qualified for further testing is for the whole period (2014-2018) or for a specific year?...
    • Line 93-96: it is not clear where these numbers come from? 2066 children were referred for testing, but in the end in 3040 a preliminary diagnosis was established based on questionnaire, testing etc?? What about the second consultation? Why a final diagnosis was made in only 2068? Where there any drop-outs? The number of allergology visits should be described in the Methods section, as well as the numbers (in the Results section)
    • Similar comments for the 16-17 age group
    • In Figure 2, the incidence of allergic rhinitis does not seem to decrease in the 7-8 years age group
    • Do the authors have an explanation for the drop in the incidence of both asthma and allergic rhinitis in 2016 in all age groups?
    • Line 129: the reference to Tab. 2 seems inappropriate

Author Response

Dear Reviewer,

We would like to thank you for your interest in our article and for the comments provided, which will undoubtedly increase the substantive value of the submitted work. We have tried to address all suggested corrections and indicated inaccuracies.

In our opinion the study sample was representative for the target population as the survey was conducted annually in all city schools and the questionnaires were distributed to all students in the surveyed age groups.

Figure 2 shows a downward trend line that was statistically significant in both age groups.

In response to the question about the drop in the incidence of both asthma and allergic rhinitis in 2016 in all age groups, we can suspect a connection with the lower concentration of particulate matter than in the previous and following years.

Sincerely,

Authors

Reviewer 2 Report

Dear authors,

I read your manuscript with great interest!

Below you will find some doubts and suggestions that I hope can contribute to the final version.

Line 43- I think that at the end of the introduction it would be important to define the aim of the study.

Line 51- How was the quality of the translation ensured? Maybe you could describe it.

Line 51- International Study of Asthma and Allergy in Childhood wouldn’t be more accurate?

Line 56- I suggest that you consider to describe how you conducted the study because some steps are only noticeable when reading the results

Line 57 - Adolescents didn’t need parental consent? Given the ethical dimension, it would be better to be clearly explained.

Line 70 and 114- The study aimed to compare the subgroups? If so, I suggest that you make it clear in your introduction (aims)

Line 75 e 76- I suggest participants instead of people.

Line 76- I don't understand the reference of “students not treated”.  It was a study aim to find it?

Line 87- I suggest forced spirometry and obstructive ventilatory impairment since it would be more accurate.

Line 89 and 105- I suggest, for a better reading, that a sentence doesn’t start with a number.

Line 130- I suggest to rewrite the paragraph to make it clearer.

Line 133- According to what I have already said, the results should be related to the aims of the study. This is the first time PM10 are mentioned.

Line 146- I suggest that results from other studies are mentioned in the introduction or discussion

Line 168- Totally agree but in your study you found that there is a reduction in incidence that can be associated with environmental measures that have been implemented, so maybe this sentence can be a little contradictory.

Line 173- Maybe “for this reason” could be eliminated.

Line 177- The reference to teachers does not seem to me to add anything to this discussion.

Line 178- I don't understand what do you mean by etiological recommendations.

Line 196- In my opinion, this figure in discussion section does not seems to add any relevant information. The text alone is clear enough.

Line 197, 198- I suggest “to be aware”. Do you have data concerning the use of face masks before de COVID-19 pandemic (I assumed that references 18 and 19 are related to air pollution as a risk factor)?

In summary, I suggest a better relationship between aims, results and discussion. Regarding the discussion, I think it should be more objective and explanations duly based on published results. I also think that it would be important to know the limitations of the study (for example, are there data on parents the smoking habits? Would it be relevant to compare your data with data, in the same period, from children living in rural areas parents?).

Best rgds

Author Response

Dear Reviewer,

We would like to thank you for your interest in our article and for the comments provided, which will undoubtedly increase the substantive value of the submitted work. We have tried to address all suggested corrections and indicated inaccuracies.

Unfortunately we don’t have data concerning the use of face masks before de COVID-19 pandemic and yes references 18 and 19 (23 and 24 in the corrected version) are related to air pollution as a risk factor.

We did not collect data concerning parents’ smoking habits.

As the data was collected by the municipal prevention program, we do not have a direct comparison with children and teenagers living in rural areas. Hence the reference to the ECAP study in the discussion.

Sincerely,

Authors

Round 2

Reviewer 1 Report

The authors provided a revised version of the manuscript.

There are only minor language issues that can be improved.

This manuscript is a resubmission of an earlier submission. The following is a list of the peer review reports and author responses from that submission.

Round 1

Reviewer 1 Report

This is an epidemiological study on the prevalence of allergic disease and asthma among children from Poland.

Please find my comments below:

Introduction:

  • the introduction does not explain with enough details the rationale and the aim for conducting such a study
  • lines 41-46 of the introduction contain redunant information that should be presented only in Methods and Results sections

Methods:

  • lines 48-49 would better fit in the Introduction section
  • the authors should give more details on how the study population was sampled. How did they ensure that the study population was a representative sample for the whole population? Why did they choose to analyse the incidence of allergies/asthma particularly in the 7-8 and 16-17 years old groups?
  • the authors should provide/explain in more detail the content of the questionnaire used for the screening of allergic disorders
  • there is no description of statistical analysis methods used

Results:

  • the authors could provide a flow chart of the final number of subjects included in the study
  • the study data should be presented in more details

Conclusion is only partially supported by the current research.

Reviewer 2 Report

Major comments:

  • This is rather a report, not an academic manuscript.
  • Lack of statistical analysis section. Need to be intensively improved
  • Only descriptive analyses were conducted. Lack of multi-variate analyses
  • Lack of significance